# A New 4-Thiazolidinone Derivative (Les-6490) as a Gut Microbiota Modulator: Antimicrobial and Prebiotic Perspectives

**DOI:** 10.3390/antibiotics13040291

**Published:** 2024-03-22

**Authors:** Yulian Konechnyi, Tetyana Rumynska, Ihor Yushyn, Serhii Holota, Vira Turkina, Mariana Ryviuk Rydel, Alicja Sękowska, Yuriy Salyha, Olena Korniychuk, Roman Lesyk

**Affiliations:** 1Department of Microbiology, Danylo Halytsky Lviv National Medical University, 69 Pekarska St., 79010 Lviv, Ukraine; tanityshka.r@ukr.net (T.R.); o_korniychuk@ukr.net (O.K.); 2Institute of Animal Biology NAAS, Vasylya Stusa St., 38, 79034 Lviv, Ukraine; yursalyha@yahoo.com; 3Department of Pharmaceutical, Organic and Bioorganic Chemistry, Danylo Halytsky Lviv National Medical University, 69 Pekarska St., 79010 Lviv, Ukraine; ihor.yushyn@gmail.com (I.Y.); golota_serg@yahoo.com (S.H.); 4Department of Organic and Pharmaceutical Chemistry, Lesya Ukrainka Volyn National University, 13 Volya Ave., 43025 Lutsk, Ukraine; 5Research Institute of Epidemiology and Hygiene, Danylo Halytsky Lviv National Medical University, 69 Pekarska St., 79010 Lviv, Ukraine; ver.apachi85@gmail.com; 6Department of Biological Chemistry, Danylo Halytsky Lviv National Medical University, 69 Pekarska St., 79010 Lviv, Ukraine; 7Department of Intellectual Property, Information and Corporate Law, Ivan Franko National University of Lviv, 1 Universytetska St., 79000 Lviv, Ukraine; maryanaryv@gmail.com; 8Department of Scientific and Medical Information and Intellectual Property, Danylo Halytsky Lviv National Medical University, 69 Pekarska St., 79010 Lviv, Ukraine; 9Microbiology Department, Ludwik Rydygier Collegium Medicum in Bydgoszcz, Nicolaus Copernicus University in Torun, 9 Maria Skłodowska-Curie St., 85-094 Bydgoszcz, Poland; asekowska@cm.umk.pl; 10Department of Biotechnology and Cell Biology, Medical College, University of Information Technology and Management in Rzeszow, Sucharskiego 2, 35-225 Rzeszow, Poland

**Keywords:** 4-thiazolidinones, antimicrobial activity, anti-inflammatory activity, metagenomic sequencing, rats, gut microbiota

## Abstract

A novel 4-thiazolidinone derivative Les-6490 (pyrazol-4-thiazolidinone hybrid) was designed, synthesized, and characterized by spectral data. The compound was screened for its antimicrobial activity against some pathogenic bacteria and fungi and showed activity against *Staphylococcus* and *Saccharomyces cerevisiae* (the Minimum Inhibitory Concentration (MIC) 820 μM). The compound was studied in the rat adjuvant arthritis model (Freund’s Adjuvant) in vivo. Parietal and fecal microbial composition using 16S rRNA metagenome sequences was checked. We employed a range of analytical techniques, including Taxonomic Profiling (Taxa Analysis), Diversity Metrics (Alpha and Beta Diversity Analysis), Multivariate Statistical Methods (Principal Coordinates Analysis, Principal Component Analysis, Non-Metric Multidimensional Scaling), Clustering Analysis (Unweighted Pair-group Method with Arithmetic Mean), and Comparative Statistical Approaches (Community Differences Analysis, Between Group Variation Analysis, Metastat Analysis). The compound significantly impacted an increasing level of anti-inflammatory microorganisms (*Blautia*, *Faecalibacterium prausnitzii*, *Succivibrionaceae*, and *Coriobacteriales*) relative recovery of fecal microbiota composition. Anti-Treponemal activity in vivo was also noted. The tested compound Les-6490 has potential prebiotic activity with an indirect anti-inflammatory effect.

## 1. Introduction

The human intestinal microbiota is a complex ecosystem that is crucial for maintaining human health. Recent studies have shown that dysbiosis, an imbalance in the microbiota, can lead to various diseases, including inflammatory bowel disease, metabolic disorders, and even rheumatic diseases [1,2,3,4,5]. Rheumatic diseases, such as rheumatoid arthritis and systemic lupus erythematosus, are chronic inflammatory diseases characterized by joint inflammation and damage to other organs, leading to pain, disability, and reduced quality of life. Despite the current treatment options, there is still a need for new therapeutic approaches that can effectively target inflammation and reduce disease activity without significant side effects.

One promising approach is the search for new anti-inflammatory agents among 4-thiazolidinone derivatives. 4-Thiazolidinone-based compounds have been shown to possess potent anti-inflammatory properties and antibacterial and antifungal activities [6,7]. In addition, these compounds are structurally diverse and can be easily modified, making them a promising class of molecules for drug development. Furthermore, their ability to modulate the gut microbiota suggests they may have additional beneficial effects beyond their anti-inflammatory properties [8].

Recent studies have suggested a link between gut microbiota and rheumatic diseases, with dysbiosis observed in patients with rheumatoid arthritis and systemic lupus erythematosus [9,10]. The mechanisms by which the gut microbiota contributes to the development of these diseases are not fully understood. Still, it is believed that dysbiosis can lead to increased intestinal permeability and the release of pro-inflammatory cytokines, leading to systemic inflammation and joint damage [11]. Therefore, identifying molecules that can modulate the gut microbiota and reduce inflammation is a promising approach to developing new therapeutics for rheumatic diseases.

In this context, this manuscript aims to synthesize a new molecule based on 4-thiazolidinone core and investigate the antimicrobial activity in vitro and anti-inflammatory activity in vivo on a model of rheumatic inflammation by studying changes in the composition of gut microbiota. 

## 2. Results

### 2.1. Chemistry

The target pyrazole-4-thiazolidinone hybrid Les-6490 **(iii)** was synthesized by Knoevenagel condensation of 1,3-diphenyl-1*H*-pyrazole-4-carbaldehyde **(i)** and thiazolidine-2,4-dione **(ii)**, with a satisfactory yield (85 %) and purity (Figure 1).

The structure of the synthesized derivative Les-6490 **(iii)** was confirmed by ^1^H, ^13^C nuclear magnetic resonance (NMR), and Liquid Chromatography–Mass Spectrometry (LC-MS) spectra (Appendix A). In the ^1^H and ^13^C NMR spectra, the signals of all the atoms appeared in the relevant magnetic field with an appropriate spectral pattern. The molecular ion peak observed at the *m/z* value of 348.0 [M + H]^+^ in the positive ionization mode in the mass spectrum confirmed the Les-6490 formation.

### 2.2. Antimicrobial Activity In Vitro and Clinical Indicators In Vivo

The studied compound Les-6490 showed activity against Gram (+) species, the reference strain *S. aureus*, and a clinical strain *Saccharomyces cerevisiae* N6 with a Minimum Inhibitory Concentration (MIC) of 820 μM (Table 1).

### 2.3. Cytotoxicity

#### 2.3.1. Cell Lines

The cytotoxic effects of Les-6490 were examined on HaCaT cell lines (spontaneously immortalized, human keratinocyte line) and lymphocytes isolated from a healthy donor. As indicated in Table 2, Les-6490 exhibited minimal impact on the viability of these pseudo-normal human cells, in contrast to doxorubicin, a well-known anticancer agent that was used as a positive control. Doxorubicin’s IC_50_ was 0.95 ± 0.20 μM for lymphocytes isolated from a healthy donor and 5.21 ± 0.86 μM for HaCaT cells, characterizing its significant effect on these cell types.

#### 2.3.2. Acute Toxicity in Mice

In the investigation of acute toxicity in mice for the newly developed compound, a lethal median dose (LD_50_) was ascertained. Freshly prepared compound solutions were administered intraperitoneally in increasing dosages ranging from 100 to 1000 mg/kg. Using the method established by Litchfield and Wilcoxon, the LD_50_ was calculated. The findings indicated that the compound exhibited relatively low acute toxicity in mice, with the LD_50_ observed to be approximately 910.0 ± 57.0 mg/kg.

### 2.4. 16S rRNA Sequence

#### 2.4.1. Composition of Microbial Community Analysis

According to the taxonomic annotation results, the top 10 taxa of each sample or group at each taxonomic rank (phylum, class, order, family, and genus) were selected to form the distribution histogram of the relative abundance of taxa to visually see the taxa with a higher relative abundance and their proportion in different classification levels of each sample. The relative abundance of taxa in the phylum is illustrated below (Appendix A). Groups 1 (AL), 2 (A), and 3 (K) have similar taxonomic compositions, representing fecal microbiota, with an abundance of *Bacteroidota*, *Firmicutes*, and *Proteobacteria* as the main phyla. Group 4 (L) additionally represents parietal (P) microbiota, including *Campilobacterota* and *Actinobacteriota* phyla.

Genus composition showed a significantly decreasing level of *Treponema* in the AL group compared to the A and K groups, which may indicate a potential anti-Treponemal activity (Figure 2). The composition of the parietal microbiota (group L) is significantly different from the fecal microbiota (groups K, A, and AL) and is characterized by a high level of *Helicobacter* genus. There are no significant (*p* < 0.05) changes between groups K, A, and AL.

According to the abundance information of the top 35 genera of all samples, the heatmap was drawn to check whether the samples with similar processing are clustered, and the similarity and difference of samples can also be observed. The result is shown in Appendix A.

#### 2.4.2. Alpha Diversity Indices

Alpha diversity is applied to the analysis of microbial community diversity within the sample [12]; analyzing the diversity of the single sample (alpha diversity) can reflect the richness and diversity of microbial communities in each sample, including species accumulation boxplot, biodiversity curves, and a series of statistical analyses. Generally speaking, OTUs generated at 97% sequence identity are considered homologous in species. Statistical indices of alpha diversity when the clustering threshold is 97% are summarized below: number of reads chosen for normalization:cutoff = 103863. The meaning of each alpha diversity index is listed in Method-Information analysis-3 alpha diversity analysis. 

The biodiversity curve shows a greater similarity between the inflammation and compound (AL) and control (K) groups than with the inflammation group (A) (Appendix A). The most diverse group was group 4 (L), representing parietal microbiota, but the most diverse group of fecal microbiota was group 2 (A).

Boxplots were formed to analyze the difference in alpha diversity indices between groups. *t*-test, Wilcox, Tukey, and TukeyHSD tests (*t*-test and Wilcox’s test are for two groups, while Wilcox, Tukey, and TukeyHSD tests are for more than two groups) were performed to analyze the significance of the difference between the groups. Boxplots based on observed species and Shannon indices are shown in Appendix A, showing the non-significant difference between all groups.

PD whole tree alpha diversity showed a significant difference between all groups (PD_whole_tree, Appendix A).

#### 2.4.3. Beta Diversity Analysis

Beta diversity represents the explicit comparison of microbial communities based on their composition. Beta diversity metrics thus assess the differences between microbial communities. To compare microbial communities between each pair of community samples, a square matrix of “distance” or “dissimilarity” was calculated to reflect the dissimilarity between certain samples, such as Unweighted Unifrac [13,14] and Weighted Unifrac distance [15]. The data in this distance matrix can be visualized using Principal Coordinate Analysis (PCoA), Principal Component Analysis (PCA), Non-Metric Multidimensional Scaling (NMDS), and Unweighted Pair-group Method with Arithmetic Means (UPGMA).

A boxplot was generated to show the difference in beta diversity indices between groups. The *t*-test, Wilcox, Tukey, and TukeyHSD tests (*t*-test and Wilcoxon tests are for two groups, while Wilcox, Tukey, and TukeyHSD tests are for groups more than two) were performed to analyze the significance of the difference between groups (Appendix A). There were no significant differences between A-AL, A-K, and K-AL groups. 

##### Principal Component Analysis (PCA) and Non-Metric Multidimensional Scaling (NMDS)

Principal Component Analysis (PCA) is a statistical procedure that extracts principle components and structures in data using orthogonal transformation and reducing data dimensionalities [16]. It extracts the first two axes, reflecting the variety of samples to the most extent. Thus, it can reflect high-dimensional data’s variation in two-dimensional graphs, revealing the simple principle embedding in complex data. Furthermore, the more similar the community composition among the samples is, the closer the distance of their corresponding data points on the PCA graph. 

The result of PCA analysis based on OTUs is shown in Figure 3. It has been shown that the K group is more similar to the AL group than the A group. The microbiota of the K and AL groups were very similar. It may represent the possible restoration of the composition of microbiota after treatment of inflammation with the tested compound. Group L is predicted to significantly differ from the other three groups because it represents the parietal microbiota.

Non-metric multidimensional scaling analysis is a ranking method applicable to ecological research [17]. It is a non-linear model designed to better represent a non-linear biological data structure, aiming to overcome the flaws in methods based on a linear model, including PCA and PCoA. The result of NMDS analysis based on OTUs is in Appendix A, and it correlates with the PCA results in Figure 3.

##### Multi-Response Permutation Procedure (MRPP) Analysis

Multi-response permutation procedure (MRPP) analysis is similar to Anosim, which aims to determine whether the difference in microbial community structures among groups is significant. It is usually applied with dimension reduction methods like PCA, PCoA, and NMDS. 

The result is shown in Table 3. The results show a statistically significant difference between groups A and AL (*p* = 0.034), which may indirectly indicate a positive effect of the tested compound on the microbiota, as groups AL and K had no statistically significant difference (*p* = 0.118).

#### 2.4.4. Between-Group Variation Analysis of Species

A statistically significant increase in the number of bacteria of the orders *Coriobacteriales* and *Aeromonadales* under the influence of the studied compound and a previous decrease under the influence of adjuvant may indicate a beneficial effect of the studied compound on inflammation and is associated with antispasmodic action (Appendix A) [18].

There was a statistical decrease in the number of bacteria of the *Neisseriaceae* family (Appendix A) and genus (Figure 4) in group AL compared with group A, which is a good trend for bacteria that typically cause inflammation. There was also a statistical increase in the number of bacteria in the family *Succivibrionaceae*, which may indicate anti-inflammatory action (Appendix A) [18,19,20].

There was a significant increase in *Faecalibacterium prausnitzii* bacteria, which have anti-inflammatory action. However, there was also a decrease in bacteria *Lactobacillus johnsonii* and *Lactobacillus reuteri*, probiotic strains that may have anti-inflammatory effects (Figure 5) [21,22,23].

Notes to Appendix A, Figure 4 and Figure 5: The left panel is the abundance of species showing significant differences between groups. Each bar represents the mean value of the abundance in each species group, showing significant differences between groups. The right panel is the confidential interval between-group variation. The left-most part of each circle stands for the lower 95% confidential interval limit, while the right-most part is the upper limit. The center of the circle stands for the difference in the mean value. The circle’s color agrees with the group whose mean value is higher. The right-most value is the p-value of the significance test of between-group variation.

#### 2.4.5. MetaStat

Taxa with significant intra-group variation are detected via metastats, a strict statistical method based on their abundance. The significance of observed abundance differences among groups is evaluated via multiple hypothesis tests for sparsely sampled features and false discovery rates (FDRs). 

Comparing the three groups reflecting the microbiota (K, A, and AL), the q value was less than 0.05 only for bacteria for the genus *Collinsella* for groups A-L (Figure 6), although there were no statistically significant differences between these two groups (Figure 4).

## 3. Discussion 

The studied derivative Les-6490 exhibited weak antimicrobial activity in vitro against the opportunistic yeast-like fungus *Saccharomyces cerevisiae* N62 (MIC 820 μM) and against the reference Gram (+) strain of *Staphylococcus aureus subsp. aureus* ATCC 25923 (MIC 2880 μM). Such activity (MICs) is much lower than antibiotics and can be considered a weak antimicrobial effect. It is difficult to explain the compound’s relative selective effect on saccharomycetes and reference staphylococci. Compounds with a similar spectrum of antimicrobial activity in the literature were inhibitors of phosphatases Cdc25B, PTP1B, and FAP-1 despite these types of phosphatases occurring only in eukaryotic cells. Yeast cell cycle regulation and phosphatase activities involve different but functionally similar proteins for cell cycle control. Therefore, these specific mammalian targets (Cdc25B, PTP1B, and FAP-1) are absent in *S. cerevisiae*. However, the compounds of PTP1B inhibitors with antimicrobial activity are described in the literature [24,25,26].

It is important to note that Les-6490 did not exhibit antimicrobial activity against the non-pathogenic probiotic strain *Limosilactobacillus fermentum*, which may indicate a low impact of the compound on non-pathogenic microbiota. *L. fermentum* is a species of lactic acid bacteria commonly found in fermenting plants and dairy products as well as in the human gastrointestinal tract and mouth. It is known for its probiotic properties, contributing to gut microbiome health by fermenting carbohydrates and producing lactic acid. *L. fermentum* can benefit the host by improving intestinal health and potentially boosting the immune system. Research into its health benefits is ongoing, particularly concerning its application in probiotic supplements and functional foods [27].

The results of the cytotoxicity study showed minimal cytotoxic effects of Les-6490 on HaCaT cell lines and isolated lymphocytes (>100 μM) as opposed to the cytotoxic impact of doxorubicin (0.95 ± 0.20 μM; 5.21 ± 0.86 μM), providing a promising outlook for Les-6490 as a safe compound for normal non-mutated human cell lines. Furthermore, the relatively low acute toxicity of Les-6490 in mice (910.0 ± 57.0 mg/kg) supports its potential as a safe therapeutic agent. 

In our previous work, we demonstrated the effect of Les-6490 on the clinical course of arthritis in rats, showing a decrease in the volume of edema of the rats’ paws and a decrease in laboratory markers of inflammation, such as the erythrocyte sedimentation rate (ESR) and leukocytosis [28].

The 16S rRNA sequencing allowed for the analysis of Les-6490’s impact on the taxonomic diversity of bacteria (phylum, class, order, family, and genus). At the genus level, Les-6490 increased diversity in the gut microbiota (Figure 2 and Appendix A), with significant anti-Treponemal activity. MRPP analysis in Table 3 demonstrated a significant (*p* < 0.05) impact of Les-6490 on the microbiota distribution in groups A–AL (*p* = 0.048), AL–L (*p* = 0.034), A–L (*p* = 0.035), and K–L (*p*-0.03). Principal component analysis (Figure 3 and Appendix A) showed that group AL (adjuvant and compound) on the graph was closer to group K (control without adjuvant and compound) than group A (adjuvant), indicating a restoration of microbiota composition in the group with the compound compared to the group without the compound.

The mechanism of anti-Treponemal action of the compound may be associated with an indirect antimicrobial effect due to competition in the bioniche and the production of bacteriocins by probiotic strains, the number of which has increased significantly, e.g., antagonism between *Bifidobacterium* species, known probiotics and bio-preservatives, and *Treponema* spp [29]. 

Gut microbial diversity and inflammation are closely linked. A diverse gut microbiome is generally associated with good health, including a well-functioning immune system [30]. Reduced microbial diversity in the gut, on the other hand, has been linked to various inflammatory conditions, such as inflammatory bowel disease (IBD), obesity, and even mental health disorders [31,32]. A diverse microbiome regulates the immune system, helps maintain the integrity of the gut barrier, and produces metabolites that can have anti-inflammatory effects. Conversely, a less diverse gut microbiome might lead to an overactive immune response and chronic inflammation. The studied compound leads to an increase in microbial diversity (Figure 2, Figure 5 and Appendix A), which may explain its anti-inflammatory effect and the positive impact on the restoration of gut microbiota.

The 16S rRNA method is not optimal for species identification of microorganisms but is suitable for genus identification. However, some publications [33,34,35] indicate this method as possible for species identification with an accuracy of up to 97–99%, depending on the genus. The difference between genera of anti-inflammatory microorganisms is significant (e.g., the genus *Faecalibacterium* in Figure 4), despite the potential inaccuracy of 16S rRNA in species identification. On the other hand, species identification of the *Collinsella* genus cannot be reliably performed using the 16S rRNA method.

The Tukey test and the Tukey–Kramer HSD test were used to determine the statistical reliability of differences between groups with four groups, with the total results shown in Appendix A. Unfortunately, there were no significant differences between the A–AL, A–K, and K–AL groups. It was, therefore, further analyzed with the help of *t*-test analysis for separately taken taxa between the two groups. Using *t*-test analysis for intergroup differences is not the most appropriate method in the case of four groups, but Metastat Analysis between groups (Figure 6) and clinical changes support our hypothesis.

There was a significant increase in *Faecalibacterium prausnitzii* (Figure 4 and Figure 5), which are anti-inflammatory commensals used as probiotic strains to treat Crohn’s disease and other inflammatory diseases [36,37] and as a probiotic/symbiotic in cancer immunotherapy [38]. There was also a statistically significant increase in the number of bacteria of the genus *Blautia* in group AL compared with group A. Bacteria of this genus can produce butyrate, which has anti-inflammatory actions and is used to treat inflammatory bowel disease [39,40,41]. In the AL group, the number of the genus Lachnospiraceae significantly increased compared with the control and group A, indicating the promotion of *Lachnospiraceae* proliferation by the study compound. *Lachnospiraceae*, a butyrate-producing bacteria beneficial for the intestinal barrier, was involved in the formation of visceral hypersensitivity [42,43]. The other genus *Collinsella* was significantly reduced in group A compared to the control, but there were no significant changes compared to group AL (Figure 4). 

There was a significant increase in the number of bacteria of the genus *Anaerobiospirillum* in group AL compared with the control, with no significant increase in group A. *Anaerobiospirillum* has been associated with health risks in humans, such as diarrhea, bacteremia, and other inflammatory responses, with a strong association with liver function [44,45].

*Colinsella*, as a part of the *Coriobacteriaceae* family, is a pro-inflammatory microorganism that can cause *Colinsella*-induced inflammation and, over time, affect glucose and obesity levels [46,47,48]. *Collinsella* is involved in the pathogenesis of rheumatoid arthritis and is a typical microorganism of the gut microbiota in patients with rheumatoid arthritis [49,50]. We expected to see an increase in the number of *Collinsella* bacteria in group A (adjuvant), but the number of bacteria was not significantly lower than in the group of healthy rats (K) in Figure 6. In contrast, the number of bacteria of the genus *Collinsella* was significantly increased in the AL group (adjuvant and compound), which is atypical, difficult to explain, and did not correlate with a significant increase in anti-inflammatory microorganisms, as well as with the clinical picture. The argument against the role of *Collinsella* as a pro-inflammatory agent is the significant increase in *Collinsella* and *Coriobacteriales* in group K compared to group A (Figure 4 and Appendix A).

Recent research indicates that not all species of the genus *Collinsella* are pro-inflammatory; some strains in the species *Collinsella aerofaciens* could be anti-inflammatory butyric acid-producing microorganisms [51] and, in some cases, modulate the anticancer immune response [52]. We hypothesize that the observed increase of *Collinsella* in the AL group may be due to the mechanism of butyrate production; however, there is currently no method available to identify *Collinsella* to the species level accurately.

Our study has several limitations, such as not all samples from the 24 rats were analyzed using 16S rRNA and that levels of pro- and anti-inflammatory cytokines, as well as specific inflammatory markers, were not measured.

Future research should focus on studying changes in the parietal microbiota under the influence of Les-6490, as well as the anti-Treponemal activity with potential connection to probiotics, the detection of anti-inflammatory products, such as butyrate, in the intestines of rats, as well as inflammatory markers and cytokines, and elucidating the exact mechanism of the in vitro antimicrobial activity by molecular docking.

## 4. Materials and Methods

### 4.1. Chemistry

#### 4.1.1. General Remarks

Melting points were measured in open capillary tubes on a BÜCHI B-545 melting point apparatus (BÜCHI Labortechnik AG, Flawil, Switzerland) and were uncorrected. Elemental analyses (C, H, N) were performed using the Perkin-Elmer 2400 CHN analyzer (PerkinElmer, Waltham, MA, USA), with results within ±0.4% of the theoretical values. The 500 MHz ^1^H and 100 MHz ^13^C NMR spectra were recorded on a Varian Unity Plus 500 (500 MHz) spectrometer (Varian Inc., Paulo Alto, CA, USA). All spectra were recorded at room temperature, unless specified otherwise, and were referenced internally to solvent reference frequencies. Chemical shifts (δ) are presented in ppm and coupling constants (*J*) in Hz. LC-MS spectra were obtained on a Finnigan MAT INCOS-50 (Thermo Finnigan LLC, San Jose, CA, USA). The reaction mixture was monitored by thin-layer chromatography (TLC) using commercial glass-backed TLC plates (Merck Kieselgel 60 F254, Merck, Darmstadt, Germany). Solvents and reagents that are commercially available were used without further purification. The synthesis of 1,3-diphenyl-1*H*-pyrazole-4-carbaldehyde **(i)** and thiazolidine-2,4-dione **(ii)** was carried out according to methods described in references [53,54], respectively.

#### 4.1.2. General Procedure for the Synthesis of 5-(1,3-diphenyl-1H-pyrazol-4-ylmethylene)-thiazolidine-2,4-dione Les-6490 (iii)

A mixture of 0.011 mol of 1,3-diphenyl-1*H*-pyrazole-4-carbaldehyde **(i)**, 0.01 mol of thiazolidine-2,4-dione **(ii)**, and 0.015 mol of ammonium acetate in 20 ml of toluene was heated under reflux for 5 h. The resultant yellow crystalline precipitate was filtered, washed with hexane, and recrystallized from a mixture of DMF–ethanol (1:2). 

Yield: 85%, yellow crystal powder, mp 278–280 °C (DMF-EtOH (1:2)). ^1^H NMR (400 MHz, DMSO-*d*_6_): δ (ppm) 7.40 (t, *J* = 7.4 Hz, 1H arom.), 7. 47–7. 59 (m, 6H, arom. + CH ylidene), 7.63 (d, *J* = 7.3 Hz, 2H arom.), 8.00 (d, *J* = 8.0 Hz, 2H, arom.), 8.68 (s, 1H, CH, pyrazole), 12.52 (s, 1H, NH, thiazolidinone). ^13^C NMR (101 MHz, DMSO-*d*_6_): δ (ppm) 115.9, 119.8, 122.5, 123.1, 127.9, 128.4, 129.2, 129.4, 129.5, 130.1, 131.8, 139.3, 154.4, 167.5 (C=O), 167.9 (C=O). LCMS (ESI+) *m/z* 348.0 (100 %, [M + H]^+^). Anal. calc. for C_19_H_13_N_3_O_2_S: C, 65.69%; H, 3.77%; N, 12.10%. Found: C, 65.90%; H, 4.00%; N, 12.30%.

### 4.2. Antimicrobial Activity In Vitro

The compound’s in vitro antimicrobial efficacy was evaluated using agar diffusion and resazurin-based microdilution assays [55,56]. For agar diffusion, 100 μL of the compound (1 mg/mL) was deposited into wells in agar, and the zones of growth inhibition were measured (well diameter: 5.5 mm). Comparative controls included DMSO and antibiotics such as vancomycin, ciprofloxacin, and clotrimazole. The resazurin-based micro-dilution assay involved a concoction of nutrient medium, microbial suspension, and the compound, augmented by resazurin, in a 96-well plate configuration (50 μL of nutrient medium (Mueller–Hinton or glucose broth), 50 ml of a microbial suspension (McFarland standard 1.0), and 100 μL of tested compounds, with the addition of 15 μL of 0.02% resazurin in each well). The study tested 11 distinct microbial and fungal strains, both reference and clinical, identified via MALDI-TOF (Bruker, Bremen, Germany) and 16S rRNA gene sequencing. Clinical strains (*Klebsiella pneumoniae* 189, *Staphylococcus aureus* N 23, *Candida albicans* N67, and *Saccharomyces cerevisiae* N62) were multidrug-resistant or extensively drug-resistant with different antibiotic resistance patterns, with *Aeromonas hydrophila* N196 being non-MDR. Clinical strains were isolated from a patient with healthcare-associated infections in regional hospitals. One strain, *Limosilactobacillus fermentum,* was a wild, non-pathogenic probiotic. All tests were conducted in triplicate.

### 4.3. In Vivo Protocol

Wistar rats, sourced from Danylo Halytsky Lviv National Medical University’s vivarium, were subjected to a week-long acclimatization before experimentation. Throughout this period, the rats had unlimited access to water and standard rodent feed, housed in a climate-controlled facility with a set temperature, humidity, and light/dark cycle (temperature: 22–24 °C, humidity: 50–65%, and a 12 h light/dark cycle). The experimental protocol was designed in strict compliance with the European Convention for the Protection of Vertebrate Animals used for Experimental and Other Scientific Purposes, Council of Europe Directive 2010/63/EU on the protection of animals used for scientific purposes, and the Law of Ukraine № 3447-IV “On the Protection of Animals from Cruelty” as amended by 440-IX of 14.01.2020, ensuring adherence to ethical standards and animal welfare regulations. To induce an inflammatory response, experimental animals were injected with Freund’s adjuvant (FA) (Thermo Fisher Scientific, San Jose, CA, USA ) 0.1 ml subcutaneously in the plantar part of the hind limb [57]. FA contains components such as BCG vaccine (Bacillus Calmette–Guérin) or polysaccharides obtained from *Mycobacterium tuberculosis*, complex fatty acids (derived from lanolin), oils, and an emulsifier in the following ratio: 10 mL of FA = 5 mL anhydrous lanolin + 15 mL of petroleum jelly + 50 mg of heat-inactivated BCG vaccine. The use of FA contributes to the emergence of delayed-type hypersensitivity and the development of autoimmune processes, predominantly affecting the joints of the rats’ hind limbs. 

After the manifestation of inflammatory signs (foot enlargement, swelling, and redness) (Appendix A) on the 7th day of the experiment, the animals received intragastric administration of Les-6490, dissolved in Tween 80 via a non-traumatic probe. The dose of the compound was calculated based on a study of similar compounds [58] with reference to the known effective dose of nimesulide. Before the introduction of the compound, the animals were fasted for 12 h [59].

Rats were humanely euthanized by decapitation. Feces were collected from the large intestine (the middle of the colon) for the study of the microbiota and a part of the colon’s wall for the study of the parietal microbiota. The rats were randomly divided into 4 groups, with 6 rats per group (Table 4). A total of 24 rats were involved, and 15 samples were sent for metagenomic 16S rRNA sequencing analysis.

### 4.4. Cytotoxicity 

#### 4.4.1. Cell Lines

The HaCaT cell line and lymphocytes isolated from a clinically healthy donor (MTT test, 48 h of cell incubation) were obtained from the Institute of Molecular Biology and Genetics, National Academy of Sciences of Ukraine, Kyiv. These cells were cultured in either Dulbecco’s Modified Eagle Medium or RPMI-1640 medium, both supplemented with 10% fetal bovine serum, in line with ATCC guidelines, and kept in a 5% CO_2_ humidified atmosphere at 37 °C.

#### 4.4.2. Acute Toxicity in Mice

Determination of Acute Toxicity: The study utilized albino male mice weighing between 23 and 25 grams. Test compounds were prepared in a saline solution (0.9% sodium chloride) with a small addition of Polysorbate 80 (Tween-80®) for better solubility. These solutions were then orally administered to mice. The median lethal dose (LD50) was determined using various doses across six animals per dose. LD50 calculations were performed using the Litchfield–Wilcoxon statistical method [60,61].

### 4.5. 16S rRNA Sequencing

To analyze the microbial community in the samples, DNA was extracted and sequenced at Novogene Bioinformatics Technology Co., Ltd. (Beijing, China). This process involved generating Operational Taxonomic Units (OTUs) by clustering effective tags with 97% similarity across the samples. Sequencing was performed using the Illumina platform, producing 250 bp paired-end reads. These reads were merged and refined to acquire clean tags, from which chimeric sequences were identified and eliminated to derive effective tags for further analyses. The out construction involved gathering data, including effective tags, low-frequency tags, and tag annotations from different samples.

#### 4.5.1. Extraction of Genome DNA

Total genome DNA was extracted from samples using the CTAB/SDS method. DNA concentration and purity were assessed on 1% agarose gel. DNA was diluted to 1 ng/μL according to the concentration using sterile water.

#### 4.5.2. Amplicon Generation

Specific primers (e.g., 16S V4: 515F-806R) were used to amplify distinct regions of the 16S rRNA genes (16SV4/16SV3/16SV3-V4/16SV4-V5), incorporating barcodes for sample identification. All PCR reactions were carried out with Phusion® High-Fidelity PCR Master Mix (New England Biolabs, Ipswich, MA, USA).

#### 4.5.3. PCR Products Quantification and Qualification, Mixing, and Purification

PCR samples were mixed with 1× loading buffer containing SYBR green and subjected to electrophoresis on a 2% agarose gel. Bands within the 400 to 450 bp range were selected, pooled at equal concentrations, and purified using the Qiagen Gel Extraction Kit (Qiagen, Germany).

#### 4.5.4. Library Preparation and Sequencing

Sequencing libraries were generated using NEBNext^®^Ultra DNA Library PreKit for Illumina; index codes were added following the manufacturer’s recommendations. Library quality was assessed on the Qubit@ 2.0 Fluorometer (Thermo Scientific) and Agilent Bioanalyzer 2100 system. At last, the library was sequenced on an Illumina platform, generating 250 bp paired-end reads.

#### 4.5.5. Paired-End Read Assembly and Quality Control

In this process, paired-end reads were assigned to samples via their unique barcodes, and the barcodes and primer sequences were then trimmed. The FLASH tool (V1.2.7, http://ccb.jhu.edu/software/FLASH/, accessed on 1 January 2022) [62] was utilized for merging these reads based on overlaps, creating raw tags. These raw tags underwent a quality control process using QIIME (V1.7.0) (V1.7.0, http://qiime.org/index.html, accessed on 1 January 2022) [63], ensuring only high-quality clean tags were retained [64]. For chimera detection, the tags were aligned against the Gold database using the UCHIME algorithm (UCHIME Algorithm, http://www.drive5.com/usearch/manual/uchime_algo.html, accessed on 1 January 2022) [65], facilitating the removal of any chimeric sequences [66]. This step was crucial for obtaining the final effective tags for the study.

#### 4.5.6. OTU Clustering

Sequence analysis was performed using Uparse software (Uparse v7.0.1001, http://drive5.com/uparse/, accessed on 1 January 2022) [67]. Sequences with ≥97% similarity were assigned to the same OTUs. The representative sequence for each OTU was screened for further annotation.

#### 4.5.7. Taxonomic Annotation

The Green Gene Database (http://greengenes.lbl.gov, accessed on 1 January 2022) [68] was used based on the RDP classifier (Version 2.2, http://sourceforge.net/projects/rdp-classifier/, accessed on 1 January 2022) [69] algorithm to assign taxonomic information to each OTU representative sequence.

#### 4.5.8. Phylogenetic Relationship Construction

In order to study the phylogenetic relationship of different OTUs and the variation in dominant species across (groups), multiple sequence alignments were conducted using the MUSCLE software (Version 3.8.3, http://www.drive5.com/muscle/, accessed on 1 January 2022) [70].

#### 4.5.9. Data Normalization

OTU abundance data were normalized using a standard sequence number corresponding to the sample with the least sequences. Subsequent alpha and beta diversity analyses were performed based on this normalized output data.

#### 4.5.10. Alpha Diversity

Alpha diversity is applied in analyzing the complexity of species diversity within a single sample by employing 6 indices, including observed-species, Chao1, Shannon, Simpson, ACE, and good-coverage. These indices in our samples were calculated with QIIME (Version 1.7.0) and visualized with R software (Version 2.15.3). The *t*-test, Wilcox, Tukey, and TukeyHSD tests (*t*-test and Wilcoxon tests are for 2 groups, while Wilcox, Tukey, and TukeyHSD analyses involve more than 2 groups) were performed to analyze the significance of the difference between groups.

#### 4.5.11. Beta Diversity

Beta diversity evaluates differences in species complexity between samples, using both weighted and unweighted UniFrac metrics in QIIME (Version 1.7.0). Dimensionality reduction in the complex data was achieved through principal component analysis (PCA) using the FactoMineR and ggplot2 packages in R (Version 2.15.3). Subsequently, Principal Coordinate Analysis (PCoA) was employed, transforming the weighted or unweighted UniFrac distance matrices into a set of orthogonal axes to visualize the data. The first two axes represented the most significant variation factors. PCoA results were visualized with the WGCNA, stat, and ggplot2 packages in R. Additionally, Unweighted Pair-group Method with Arithmetic Means (UPGMA) Clustering was performed using QIIME for hierarchical clustering to interpret the distance matrix with average linkage. The *t*-test, Wilcox, Tukey, and TukeyHSD tests (*t*-test and Wilcoxon tests are for 2 groups, while Wilcox, Tukey, and TukeyHSD analyses involve more than 2 groups) were performed to analyze the significance of the difference between groups.

## 5. Conclusions

The new 4-thiazolidininone derivative, Les-6490, was synthesized and exhibited antimicrobial activity against Gram (+) microorganisms in vitro and anti-*Treponemal* activity in vivo. Additionally, Les-6490 increased the abundance of anti-inflammatory microorganisms (*Blautia, Faecalibacterium prausnitzii, Succivibrionaceae,* and *Coriobacteriales*) in vivo, suggesting potential prebiotic and anti-inflammatory properties of the molecule.

## 6. Patents

An electronic application for obtaining a patent for an invention was submitted with application No. 280013 on 31 January 2024, to the Ministry of Economy of Ukraine, National Intellectual Property Authority, State Organization “Ukrainian National Office of Intellectual Property and Innovation” (UKRNOIVI).

## Figures and Tables

**Figure 1 antibiotics-13-00291-f001:**
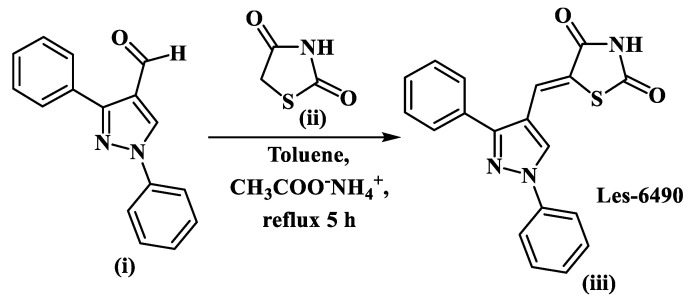
Scheme of 5−(1,3−diphenyl−1*H*−pyrazol−4−ylmethylene)−thiazolidine−2,4−dione (iii, Les−6490) synthesis.

**Figure 2 antibiotics-13-00291-f002:**
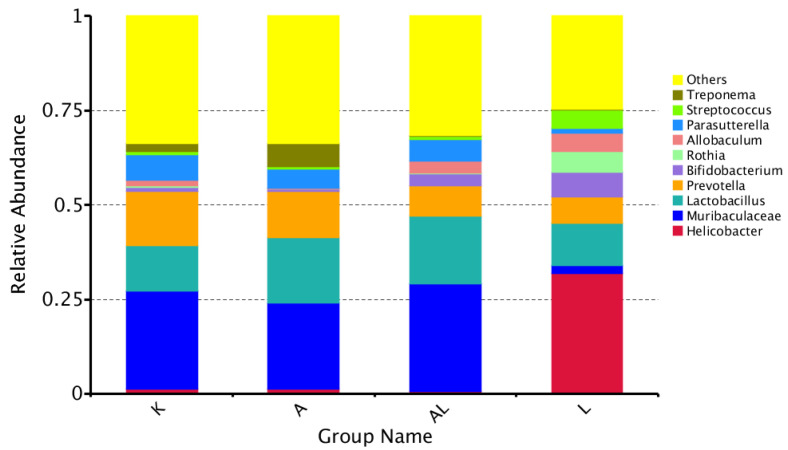
Taxonomic relative abundance at the genus level across different groups. This bar chart displays the genus-level distribution and relative abundance of microbial taxa. It showcases key genera within the fecal microbiota of groups K (control without adjuvant and compound), A (adjuvant), AL (adjuvant and compound), and L (compound without adjuvant).

**Figure 3 antibiotics-13-00291-f003:**
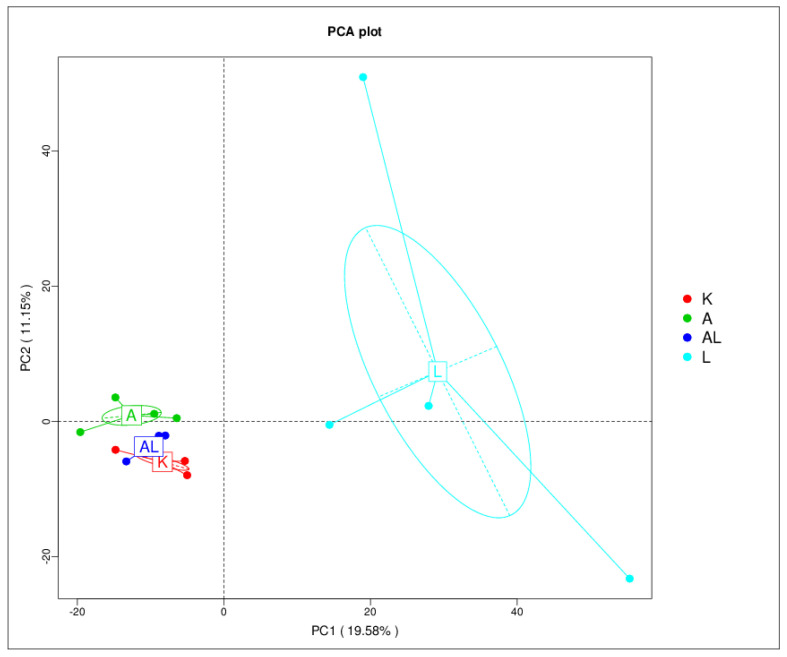
Principal component analysis (PCA) with cluster distribution. This figure shows a PCA plot illustrating the distribution of microbial communities across the study groups: K (control without adjuvant and compound), A (adjuvant), AL (adjuvant and compound), and L (compound without adjuvant). The plot reveals distinct clustering patterns, indicating differences in microbial composition among the groups.

**Figure 4 antibiotics-13-00291-f004:**
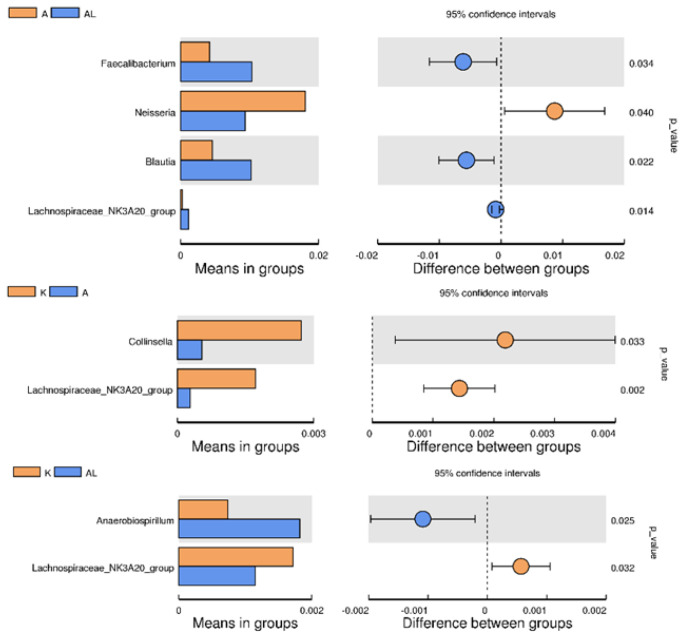
Between−group T−test analysis (genus *Faecalibacterium*, genus *Neisseria*, genus *Blautia*, genus *Lachnospiraceae* NK3A20 group, genus *Colinsella*, and genus *Anaerobiospirilum*) across the study groups: K (control without adjuvant and compound), A (adjuvant), AL (adjuvant and compound), and L (compound without adjuvant). The analysis aims to highlight significant variations in the abundance of these genera among the groups, providing insights into their respective roles in the microbial ecosystem under different treatment conditions.

**Figure 5 antibiotics-13-00291-f005:**
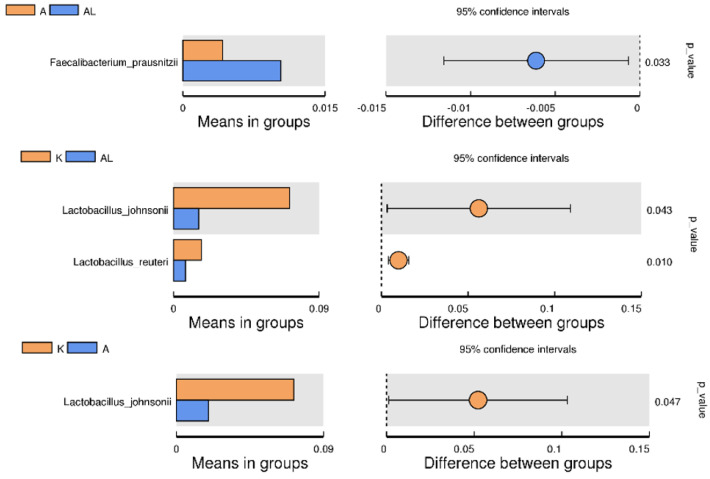
Between−group T−test analysis (species Faecalibacterium prausnitzii, species Lactobacillus johnsonii, and Lactobacillus reuteri) across the groups: K (control without adjuvant and compound), A (adjuvant), AL (adjuvant and compound), and L (compound without adjuvant). The analysis aims to highlight how these specific species vary in response to different treatment conditions, offering insights into their roles within the microbial ecosystem.

**Figure 6 antibiotics-13-00291-f006:**
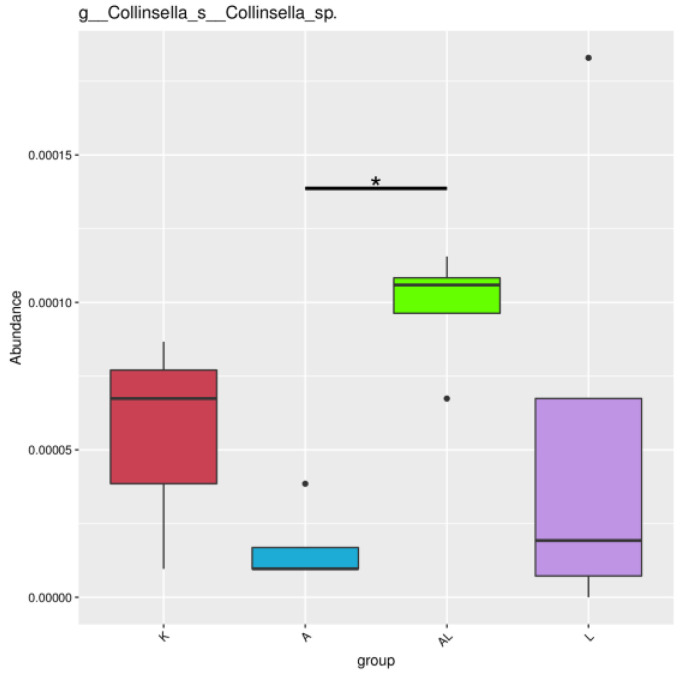
Metastat analysis between groups. Notes: Plotted by relative abundance on the Y-axis and the group name on the X-axis. The horizontal line represents the two groups with significant variation. “*” represents significant variation (q value < 0.05). Groups: K (control without adjuvant and compound), A (adjuvant), AL (adjuvant and compound), and L (compound without adjuvant).

**Table 1 antibiotics-13-00291-t001:** In vitro antimicrobial activity of Les-6490 (zone of growth inhibition at conc. 1 mg/mL after 24–48 h and MIC value μM).

N	Type of Species		Species of Bacteria and Fungi	Zone of Growth Inhibition (mm ± SE) and Some MIC Value (μM)				
				Les-6490	DMSO	Vanco- Mycin	Cipro- Floxacin	Clotri- MAZOLE
1	Gram-negative bacteria	Reference strains	*Pseudomonas aeruginosa ATCC 10145*	00	7.0 ± 0.3	-	35.0 ± 0.3	-
2			*Raoultella terrigena* ATCC 33257	00	6.5 ± 0.25	-	42.0 ± 0.5	-
3		Clinical strains	*Klebsiella pneumoniae* 189	00	n/a	-	20.0 ± 0.2	-
4			*Aeromonas hydrophila* N196	00	n/a	-	27.0 ± 0.4	-
5		Wild non-pathogenic probiotic strain	*Limosilactobacillus fermentum*	00	n/a	-	43.0 ± 0.5	-
6	Gram-positive bacteria	Reference strains	*Streptococcus agalactiae* ATCC 13813	00	n/a	32.0 ± 0.5	-	-
7			*Staphylococcus aureus subsp. aureus* ATCC 25923	15.4 ± 0.4 MIC 2880 μM	n/a	32.0 ± 0.5	35.0 ± 0.5	-
8		Clinical strains	*Staphylococcus aureus* N 23	00	n/a	11.4 ± 0.3	9.0 ± 0.2	-
9	Fungi	Reference strains	*Candida. albicans* (ATCC 885-653)	20.0 ± 0.4	n/a	-	-	18.0 ± 0.5
10		Clinical strains	*Candida albicans* N67	22.0 ± 0.3	n/a	-	-	11.0 ± 0.3 MIC 2.9 µM
11			*Saccharomyces cerevisiae* N62	25.0 ± 0.4 MIC 820 μM	n/a	-	-	8.0 ± 0.2

Vancomycin 30 µg (inhibition zone 17–21 mm for *S. aureus*); ciprofloxacin 5 µg (inhibition zone 25–33 mm for *P. aeruginosa*, 22–30 mm for *S. aureus*); clotrimazole 10 µg (inhibition zone 12–17 mm for *Candida* spp.; diameter of well 5.5 mm; n/a no activity; ‘-’ not tested.

**Table 2 antibiotics-13-00291-t002:** IC_50_ value of Les-6490 for a panel of human cell lines (MTT test, 48 h, m±SD, and μM).

Comp./Cell Line	Isolated Lymphocytes	HaCaT
Les-6490	>100	>100
Doxorubicin	0.95 ± 0.20	5.21 ± 0.86

**Table 3 antibiotics-13-00291-t003:** MRPP results.

Type of Microbiota	Group	A	Observed-Delta	Expected-Delta	Significance
Fecal–fecal	A-K	0.06746	0.4858	0.521	0.085
Fecal–fecal	A-AL	0.05647	0.4808	0.5095	0.048
Fecal–fecal	AL-K	0.05719	0.4668	0.4951	0.118
Fecal–parietal	AL-L	0.2289	0.4949	0.6419	0.034
Fecal–parietal	A-L	0.2042	0.5116	0.6428	0.035
Fecal–parietal	K-L	0.1988	0.502	0.6266	0.03

Note: A small value in the column titled observed-delta indicates that the inner-group variation is small, while a large one in the column of expected-delta means that the inter-group variation is great. A positive A-value suggests that variation among groups is larger than variation within groups, while a negative one shows the opposite relationship. The difference among groups is significant if the value in the column of significance is less than 0.05. Similar results were noted using ADONIS (permutational MANOVA or nonparametric MANOVA) and analysis of molecular variance (AMOVA) approaches.

**Table 4 antibiotics-13-00291-t004:** Experimental groups of rats.

Group N	Group Name	Group Description	N of Samples (n) for 16S rRNA	Type of Studied Microbiota
1	A+L	Adjuvant (A) and compound (L)	4	F
2	A	Adjuvant (A)	4	F
3	K	Control without adjuvant and compound (K)	3	F
4	L	Compound without adjuvant (L)	4	P
Total			11	F
			4	P
			15	

F: fecal microbiota; P: parietal microbiota.

## Data Availability

Data are contained within the article and Appendix A.

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
