# Peer review of "A New 4-Thiazolidinone Derivative (Les-6490) as a Gut Microbiota Modulator: Antimicrobial and Prebiotic Perspectives"

_antibiotics, 2024, doi:10.3390/antibiotics13040291_

Round 1
Reviewer 1 Report
Comments and Suggestions for Authors|
Title: Composition of rats’ gut microbiota under the influence of new 4-thiazolidinone derivative (Les-6490), as a potential prebiotic/antimicrobial agent |
|
|
|
|
Authors:
Yulian Konechnyi, Tetyana Rumynska, Ihor Yushyn, Serhii Holota, Vira Turkina, Mariana Ryviuk Rydel, Alicja Sękowska, Yuriy Salyha, Olena Korniychuk and Roman Lesyk
The paper focused on synthesizing a new molecule based on 4-thiazolidinone core, its investigation for antimicrobial activity in vitro and anti-inflammatory activity in vivo on a model of rheumatic inflammation by studying changes in the composition of gut microbiota.
(1) Page 1 Abstract
“The Compound was studied in the rat adjuvant arthritis model (Freund’s Adjuvant) in vivo.”
Figure S4. The photo of rat paw after the introduction of Freund's adjuvant (AF) shows signs of inflammation, swelling, and redness.
Please provided the photo of rat paw after the treatment by Adjuvant (A) + compound (L) , Control without adjuvant and com-pound (K) and compound without adjuvant (L) .
(2) Page 4
“Doxorubicin’s IC50 ranged from 0.95±0.20 to 5.21±0.86 μM ”
The expression above is incorrect. IC50 value should not be in a range from 0.95±0.20 to 5.21±0.86 μM. You should separate state doxorubicin’s IC50 on lymphocytes isolated and HaCaT cell lines.
Comments on the Quality of English LanguageModerate editing of English language required
Author Response
Reviewer: 1
Dear reviewer!
Many thanks for Your time spending and efforts in reviewing the manuscript. All changes are highlighted in gray.
Point-by-point response to Comments and Suggestions for Authors
Comment 1: (1) Page 1 Abstract
“The Compound was studied in the rat adjuvant arthritis model (Freund’s Adjuvant) in vivo.”
Figure S4. The photo of rat paw after the introduction of Freund's adjuvant (AF) shows signs of inflammation, swelling, and redness.
Please provided the photo of rat paw after the treatment by Adjuvant (A) + compound (L) , Control without adjuvant and com-pound (K) and compound without adjuvant (L) .
Response 1: Photo of paws added.
Comment 2: (2) Page 4
“Doxorubicin’s IC50 ranged from 0.95±0.20 to 5.21±0.86 μM ”
The expression above is incorrect. IC50 value should not be in a range from 0.95±0.20 to 5.21±0.86 μM. You should separate state doxorubicin’s IC50 on lymphocytes isolated and HaCaT cell lines.
Response 2: The sentence has been rephrased, thanks for the suggestion.
Sincerely,
Yulian Konechnyi

Reviewer 2 Report
Comments and Suggestions for Authors
This manuscript describes the synthesis and characterization of Les-6490, a novel 4-thiazolidinone derivative, with demonstrated antimicrobial activity against Staphylococcus and Saccharomyces cerevisiae. In vivo studies using a rat adjuvant arthritis model showed its ability to modulate parietal and fecal microbial composition, notably increasing levels of anti-inflammatory microorganisms like Blautia and Faecalibacterium prausnitzii. The compound also displayed potential prebiotic activity and indirect anti-inflammatory effects, highlighting its promising therapeutic potential in treating rheumatic inflammation through gut microbiota modulation. Overall, while the manuscript presents valuable findings, addressing the following suggestions would improve its scientific rigor and clarity, making it more suitable for publication.
1. The antimicrobial activity of Les-6490 against Saccharomyces cerevisiae and Staphylococcus aureus, while reported, is relatively weak compared to standard antibiotics. Further discussion on the mechanism underlying this activity and comparison with existing literature would enhance the understanding of its efficacy.
2. The discussion on the compound's impact on gut microbiota diversity and its potential anti-inflammatory properties is insightful. However, the study lacks comprehensive analysis of specific inflammatory markers and cytokines, which could provide more concrete evidence of its anti-inflammatory effects.
3. The study acknowledges several limitations, including the small sample size for 16S rRNA analysis and the absence of measurement for specific inflammatory markers. Addressing these limitations and providing suggestions for future research could strengthen the manuscript.
Author Response
Reviewer: 2
Dear reviewer!
Many thanks for Your time spending and efforts in reviewing the manuscript. All changes are highlighted in gray.
Point-by-point response to Comments and Suggestions for Authors
Comment 1: The antimicrobial activity of Les-6490 against Saccharomyces cerevisiae and Staphylococcus aureus, while reported, is relatively weak compared to standard antibiotics. Further discussion on the mechanism underlying this activity and comparison with existing literature would enhance the understanding of its efficacy.
Response 1: We acknowledge that the antimicrobial activity of Les-6490 against Saccharomyces cerevisiae and Staphylococcus aureus is comparatively weaker than standard antibiotics. Added about it in the discussion. In the discussion section, there are speculations about the mechanism of action of the molecule, but further research is needed.
Comment 2: The discussion on the compound's impact on gut microbiota diversity and its potential anti-inflammatory properties is insightful. However, the study lacks comprehensive analysis of specific inflammatory markers and cytokines, which could provide more concrete evidence of its anti-inflammatory effects.
Response 2: We acknowledge the reviewer's point regarding the need for a more comprehensive analysis of specific inflammatory markers and cytokines to provide concrete evidence of Les-6490's anti-inflammatory effects. Our study indeed focused on the broader impact of Les-6490 on gut microbiota diversity and inferred anti-inflammatory properties based on changes in microbial composition known to influence inflammation. We plan to incorporate detailed analysis of key inflammatory markers and cytokines in future work to elucidate the compound's specific anti-inflammatory mechanisms further.
To be honest, we planned to check IL-1,2,6 and cyclooxygenase levels, and we planned to do it in March 2022, but we were only able to do it in June 2022 (due to political situation), and no changes were detected in the serum, probably the serum was already unusable 6 mouth after experiment.
Comment 3: The study acknowledges several limitations, including the small sample size for 16S rRNA analysis and the absence of measurement for specific inflammatory markers. Addressing these limitations and providing suggestions for future research could strengthen the manuscript.
Response 3: Added as suggested.
Sincerely,
Yulian Konechnyi

Reviewer 3 Report
Comments and Suggestions for Authors
This paper provides interesting data. However, further improvement is necessary before publication.
1. In Fig 11-14, these results should be analyzed for the significant difference between the 2 groups by multiple-range test such as Tukey test, Tukey-Kramer HSD test, etc., but not by T-test because this study was conducted with 4 groups. Be cautious when T-test was used for the analysis between the 2 groups because generally scientists tend to choose convenient groups (for the authors hypothesis; Cherry-pick data) indicating the significant difference between the 2 groups.
2. Fig 15 appears to indicate the significant increase of Collinsella
abundance in AL group (Adjuvant + test compound Les-6490) compared to A group (Adjuvant). According to the references indicating pro-inflammatory role of Collinsella, the test compound may increase a harmful microorganism. This result is critical for evaluating the hypothesis that this compound (Les-6490) is a prebiotic.
3. In the data of Fig 3, authors must make clear to show whether the test compound statistically increased the abundance of Bifidobacterium and Lactobacillus, the most typical probiotics in the A and AL groups.
4. This study does not indicate the results of Succivibionaceae and Coriobacteriales (Line 39 of Abstract) in the comparison of the A & AL groups. Are these microorganisms beneficial or probiotics (How about the references ?)?
5. How about the modulations of short-chain fatty acids by Les-6490?
6. Authors should reconsider the title “Composition of rats’ gut microbiota under the influence of new 4-thiazolidinone derivative (Les-6490), as a potential prebiotic/antimicrobial agent”. This study does not provide enough evidence indicating that this compound is a prebiotic.
7. The rationale for the dose of Les-6490 is necessary in the study of gut microflora in the mice experiment.
Author Response
Reviewer: 3
Dear reviewer!
Many thanks for Your time spending and efforts in reviewing the manuscript. All changes are highlighted in gray.
Point-by-point response to Comments and Suggestions for Authors
Comment 1: In Fig 11-14, these results should be analyzed for the significant difference between the 2 groups by multiple-range test such as Tukey test, Tukey-Kramer HSD test, etc., but not by T-test because this study was conducted with 4 groups. Be cautious when T-test was used for the analysis between the 2 groups because generally scientists tend to choose convenient groups (for the authors hypothesis; Cherry-pick data) indicating the significant difference between the 2 groups.
Response 1: We used Tukey test and Tukey-Kramer HSD test, the total results are shown in Figure 6 and Figure 8. Unfortunately, in total for all types of microorganisms, there were no significant differences between A-AL, A-K, and K-AL groups, so it was analyzed for separate classification units (species, genus, etc.). Added relevant clarifications to Results, Discussion, Methods.
Yes, we agree that the emphasis on T-test analysis between group differences can appear as Cherry-pick data, but Metastat Analysis between groups (Figure 15) confirms our assumption, as well as the clinical improvement decrease in the volume of edema of the rats' paws and a decrease in laboratory markers of inflammation such as erythrocyte sedimentation rate (ESR), and leukocytosis [27].
Comment 2: Fig 15 appears to indicate the significant increase of Collinsella abundance in AL group (Adjuvant + test compound Les-6490) compared to A group (Adjuvant). According to the references indicating pro-inflammatory role of Collinsella, the test compound may increase a harmful microorganism. This result is critical for evaluating the hypothesis that this compound (Les-6490) is a prebiotic.
Response 2: Thank you for highlighting the findings presented in Figure 15 regarding the increase of Collinsella abundance in the AL group compared to the A group. We have added the relevant rationale to the discussion.
Collinsella species are indeed often associated with pro-inflammatory states in the literature; however, the relationship between specific microbial taxa and their impact on host health is complex and context-dependent. The observed increase in Collinsella abundance in the context of Les-6490 administration warrants a nuanced interpretation.
While some studies have linked Collinsella to pro-inflammatory conditions, its role is not universally detrimental across all contexts. The significance of its increased abundance in the AL group may also reflect a transitional state in the gut microbiota's adaptation to Les-6490, potentially contributing to a more diverse microbial ecosystem. Furthermore, the overall impact of Les-6490 on gut health and inflammation cannot be solely determined by the presence of a single microbial genus but should be considered within the broader context of microbiota dynamics and host response, including analysis of inflammatory markers and cytokines, which we aim to address in future studies.
This finding underscores the importance of conducting comprehensive studies to elucidate the multifaceted roles of gut microbiota in health and disease. We acknowledge the need for further investigation into the implications of increased Collinsella abundance and its mechanistic pathways in the context of Les-6490's prebiotic potential.
Comment 3: In the data of Fig 3, authors must make clear to show whether the test compound statistically increased the abundance of Bifidobacterium and Lactobacillus, the most typical probiotics in the A and AL groups.
Response 3: Thank you for the valid comment. We checked again. There are no significant changes between groups K, A, AL in Figure 3. Our conclusion is not based on this Figure, because Figure 3 shows the 10 most quantitatively represented genera, and anti-inflammatory bacteria (Blautia, Faecalibacterium prausnitzii, Succivibionaceae, Coriobacteriales) are not among these genera.
Comment 4: This study does not indicate the results of Succivibionaceae and Coriobacteriales (Line 39 of Abstract) in the comparison of the A & AL groups. Are these microorganisms beneficial or probiotics (How about the references ?)?
Response 4: The letter "r" in the word "Succivibrionaceae" was omitted. The error has been fixed. In Figure 12 and references 18-20 about this.
Comment 5: How about the modulations of short-chain fatty acids by Les-6490?
Response 5: Since, during the experiment, we did not foresee an increase in the number of microorganisms that produce SCFAs, therefore this study was not planned. But we must take these studies into account when planning the next experiments, as they are valuable for understanding the mechanisms of action of new compounds.
Comment 6: Authors should reconsider the title “Composition of rats’ gut microbiota under the influence of new 4-thiazolidinone derivative (Les-6490), as a potential prebiotic/antimicrobial agent”. This study does not provide enough evidence indicating that this compound is a prebiotic.
Response 6: We partially agree, we believe that an indirect prebiotic effect is present, through the selective induction of anti-inflammatory microorganisms. But we also agree that this activity is not sufficiently well studied. Corrected name of the manuscript to: «A new 4-thiazolidinone derivative (Les-6490) as a gut microbiota modulator: antimicrobial and prebiotic perspectives»
Comment 7: The rationale for the dose of Les-6490 is necessary in the study of gut microflora in the mice experiment.
Response 7: For this purpose, previous studies of similar compounds [DOI: 10.7324/JAPS.2017.70129] were taken into account. Nimesulide, for which an effective dose is established, was also used as a comparison drug. Our compound should have the same or greater efficacy at the same dose. That is, a guideline based on a known dose of the comparison drug. Added in Methods.
Sincerely,
Yulian Konechnyi

Reviewer 4 Report
Comments and Suggestions for Authors
The manuscript titled "Composition of rats’ gut microbiota under the influence of new 4-thiazolidinone derivative (Les-6490) as a potential prebiotic/antimicrobial agent" presents an investigation into the effects of a novel 4-thiazolidinone derivative, Les-6490, on the gut microbiota composition in a rat model. This study aims to explore the potential of Les-6490 as both a prebiotic and antimicrobial agent, with a focus on its impact on anti-inflammatory microorganisms and its ability to modulate gut microbiota composition.
Some posititive and negative aspects of the study are as follows:
Innovative Approach: The study introduces a novel compound, Les-6490, and explores its dual potential as a prebiotic and antimicrobial agent, which is a significant contribution to the field.
Comprehensive Analysis: Utilizes a range of analytical techniques (e.g., 16S rRNA metagenome sequencing, diversity metrics, multivariate statistical methods) to assess the impact of Les-6490 on gut microbiota.
In Vivo Validation: Conducts an in-depth examination of Les-6490's effects in a rat model, providing valuable insights into its potential therapeutic applications.
Limited Scope of Microbial Analysis: While the study examines the impact on gut microbiota, it might benefit from a broader analysis, including potential effects on other body systems or microbiomes.
Specificity of Findings: The results are based on a specific animal model (rats), which may limit the generalizability of the findings to humans without further research.
Lack of Comparative Analysis: The study could be strengthened by comparing the effects of Les-6490 with those of existing prebiotic and antimicrobial agents to contextualize its efficacy.
This study presents a promising exploration of Les-6490's potential as a novel therapeutic agent. It generally follows a sound scientific methodology, with a clear structure in its methods, results, and conclusions. However, a few potential areas for improvement or considerations that could be seen as flaws include:
Methods Section:
- The description of the experimental design and statistical analysis could be more detailed to enhance reproducibility. For example, specifying the software or statistical tests used could clarify the analysis process.
- While the animal model is appropriate, the manuscript could benefit from a justification of the choice of the rat model over other models, explaining why it's most suitable for studying the effects of Les-6490.
-
Discussion of Limitations:
-
A more thorough discussion of the study's limitations, including the potential impact of the animal model's specificities on the generalizability of the findings and any limitations in the experimental design, would provide a more balanced view of the research.
-
Broader Contextualization in Conclusion:
-
The conclusion section could be improved by more explicitly linking the study's findings to the broader context of gut microbiota research, including how Les-6490 fits into existing prebiotic and antimicrobial strategies and potential implications for future research directions.
Addressing these points could enhance the manuscript's clarity, robustness, and impact, ensuring that its contributions to the field are well-articulated and understood within the context of existing research, making a stronger case for Les-6490's potential as a novel prebiotic/antimicrobial agent.
Author Response
Reviewer: 4
Dear reviewer!
Many thanks for Your time spending and efforts in reviewing the manuscript. All changes are highlighted in gray.
Point-by-point response to Comments and Suggestions for Authors
Comment 1: Innovative Approach: The study introduces a novel compound, Les-6490, and explores its dual potential as a prebiotic and antimicrobial agent, which is a significant contribution to the field.
Comprehensive Analysis: Utilizes a range of analytical techniques (e.g., 16S rRNA metagenome sequencing, diversity metrics, multivariate statistical methods) to assess the impact of Les-6490 on gut microbiota.
In Vivo Validation: Conducts an in-depth examination of Les-6490's effects in a rat model, providing valuable insights into its potential therapeutic applications.
Response 1: Thank you for acknowledging the innovative approach, comprehensive analysis, and in vivo validation in our study of Les-6490. We are grateful for your recognition of the work's contribution to the field. The dual potential of Les-6490 as both a prebiotic and antimicrobial agent indeed represents a promising avenue for future therapeutic applications. Our utilization of diverse analytical techniques and in-depth examination in a rat model aimed to provide a thorough understanding of its effects. We are committed to continuing this research to further elucidate Les-6490's potential benefits and mechanisms of action.
Comment 2: Limited Scope of Microbial Analysis: While the study examines the impact on gut microbiota, it might benefit from a broader analysis, including potential effects on other body systems or microbiomes.
Specificity of Findings: The results are based on a specific animal model (rats), which may limit the generalizability of the findings to humans without further research.
Lack of Comparative Analysis: The study could be strengthened by comparing the effects of Les-6490 with those of existing prebiotic and antimicrobial agents to contextualize its efficacy.
Response 2: Limited Scope of Microbial Analysis: We appreciate the suggestion to expand our microbial analysis. Future studies will aim to explore Les-6490's effects beyond the gut microbiota, including its impact on other body systems and microbiomes, to provide a holistic view of its therapeutic potential.
Specificity of Findings: We acknowledge the limitations of using a specific animal model for generalizing findings to humans. Further research, including human clinical trials, will be essential to validate Les-6490's efficacy and safety in human populations.
Lack of Comparative Analysis: The recommendation to include comparative analyses with existing prebiotic and antimicrobial agents is well-taken. Such comparisons in future work will help to better contextualize Les-6490’s efficacy, enhancing our understanding of its place within the broader landscape of antimicrobial and prebiotic therapies.
Comment 3: This study presents a promising exploration of Les-6490's potential as a novel therapeutic agent. It generally follows a sound scientific methodology, with a clear structure in its methods, results, and conclusions. However, a few potential areas for improvement or considerations that could be seen as flaws include:
Methods Section: The description of the experimental design and statistical analysis could be more detailed to enhance reproducibility. For example, specifying the software or statistical tests used could clarify the analysis process.
While the animal model is appropriate, the manuscript could benefit from a justification of the choice of the rat model over other models, explaining why it's most suitable for studying the effects of Les-6490.
Response 3: Rheumatoid arthritis (RA) in rats reproduces the clinical and pathological features of RA in humans. This means that studying the gut microbiome in rats with RA can provide important insights into the mechanisms underlying this disease in humans. The rat model of RA is an important tool for studying the gut microbiota, as it allows studying the relationship between the microbiota and the development of the disease, as well as the influence of drugs and other factors on this process [https://doi.org/10.1093/rap/rkad034]. We appreciate your feedback on the methods section.
Comment 4: Discussion of Limitations: A more thorough discussion of the study's limitations, including the potential impact of the animal model's specificities on the generalizability of the findings and any limitations in the experimental design, would provide a more balanced view of the research.
Response 4: We acknowledge the importance of a balanced discussion on limitations. The final manuscript include an expanded section addressing the limitations related to the use of the rat model and its implications for generalizability, as well as any constraints in the experimental setup. This ensure a comprehensive understanding of the study's context and areas for future research.
Comment 5: Broader Contextualization in Conclusion:
The conclusion section could be improved by more explicitly linking the study's findings to the broader context of gut microbiota research, including how Les-6490 fits into existing prebiotic and antimicrobial strategies and potential implications for future research directions.
Addressing these points could enhance the manuscript's clarity, robustness, and impact, ensuring that its contributions to the field are well-articulated and understood within the context of existing research, making a stronger case for Les-6490's potential as a novel prebiotic/antimicrobial agent.
Response 5: We refine our conclusion to more strongly connect our findings with the wider body of gut microbiota research, emphasizing Les-6490's place among existing prebiotic and antimicrobial approaches. This enhancement clarify its contributions and potential implications, solidifying Les-6490's significance as an innovative therapeutic agent in the field.
Sincerely,
Yulian Konechnyi

Round 2
Reviewer 1 Report
Comments and Suggestions for Authors
The manuscript-antibiotics-2897908 can be accepted in present form.
Comments on the Quality of English LanguageExtensive editing of English language required
Author Response
Reviewer: 1 (Round 2)
Dear reviewer!
Many thanks for Your time spending and efforts in reviewing the manuscript. All changes are highlighted in green. The text has been checked by a certified English translator, changes are highlighted in red.
Point-by-point response to Comments and Suggestions for Authors
Comment 1: The manuscript-antibiotics-2897908 can be accepted in present form. Extensive editing of English language required
Response 1: We truly appreciate your positive feedback on the manuscript's potential and its adherence to scientific methodology. In response to your comments on English language quality, the manuscript was proofread by a certified English translator.
Sincerely,
Yulian Konechnyi

Reviewer 3 Report
Comments and Suggestions for Authors
This manuscript is well improved, and appropriate for publication.
Author Response
Reviewer: 3 (Round 2)
Dear reviewer!
Many thanks for Your time spending and efforts in reviewing the manuscript. All changes are highlighted in green. The text has been checked by a certified English translator, changes are highlighted in red.
Point-by-point response to Comments and Suggestions for Authors
Comment 1: This manuscript is well improved, and appropriate for publication.
Response 1: Thank you for your positive evaluation and recommendation for publication. We are grateful for the support and constructive feedback provided throughout the review process.
Sincerely,
Yulian Konechnyi
